# Improved Cultivation and Isolation of Diverse Endophytic Bacteria Inhabiting *Dendrobium* Roots by Using Simply Modified Agar Media

Tomoki Nishioka,[a] Hideyuki Tamaki[a,b,c]

[a]Bioproduction Research Institute, National Institute of Advanced and Industrial Science and Technology (AIST), Tsukuba, Ibaraki, Japan
[b]Faculty of Life and Environmental Sciences, University of Tsukuba, Tsukuba, Ibaraki, Japan
[c]Microbiology Research Center for Sustainability (MiCS), University of Tsukuba, Tsukuba, Ibaraki, Japan

**ABSTRACT** *Dendrobium* plants are members of the family *Orchidaceae*, many of which are endangered orchids with ornamental and medicinal values. *Dendrobium* endophytic microbes have attracted attention for the development of strategies for plant protection and utilization of medicinal principles. However, the role of endophytic bacteria is poorly elucidated due to the lack of their successful cultivation. This study obtained a total of 749 endophytic isolates from *Dendrobium* roots using solid media prepared by simply modified methods (separate sterilization of phosphate and agar [PS] and use of gellan gum as a gelling reagent [GG]) and by a conventional method of autoclaving the phosphate and agar together (PT method). Notably, based on a comparison of 16S rRNA gene sequences between the isolates and the *Dendrobium* root endophyte community, we successfully retrieved more than 50% (17 out of 30) of the predominant endophytic bacterial operational taxonomic units (OTUs) using PS and GG media, which is a much higher recovery rate than that of PT medium (16.7%). We further found that a number of recalcitrant bacteria, including phylogenetically novel isolates and members of even the rarely cultivated phyla *Acidobacteriota* and *Verrucomicrobiota*, were obtained only when using PS and/or GG medium. Intriguingly, the majority of these recalcitrant bacteria formed colonies faster on PS or GG medium than on PT medium, which may have contributed to their successful isolation. Taken together, this study succeeded in isolating a wide variety of *Dendrobium* endophytic bacteria, including predominant ones using PS and GG media, and enables performance of future studies to clarify their unknown roles associated with the growth of *Dendrobium* plants.

**IMPORTANCE** *Dendrobium* endophytic bacteria are of great interest since their functions may contribute to the protection of endangered orchids with ornamental and medicinal values. To understand and reveal the "true roles" of the endophytes, obtaining those axenic cultures is necessary even in the metagenomic era. However, no effective methods for isolating a variety of endophytic bacteria have been established. This study first demonstrated that the use of simply modified medium is quite effective and indeed allows the isolation of more than half of the predominant endophytic bacteria inhabiting *Dendrobium* roots. Besides, even phylogenetically novel and/or recalcitrant endophytic bacteria were successfully obtained by the same strategy. The obtained endophytic bacteria could serve as "living material" for elucidating their unprecedented functions related to the conservation of endangered orchid plants. Furthermore, the culture method used in this study may enable the isolation of various endophytic bacteria dominating not only in orchid plants but also in other useful plants.

**KEYWORDS** *Dendrobium*, endophytic bacteria, isolation and cultivation, novel bacteria, orchid, predominant bacteria

Address correspondence to Tomoki Nishioka, tomoki-nishioka@aist.go.jp, or Hideyuki Tamaki, tamaki-hideyuki@aist.go.jp.

The authors declare no conflict of interest.

*O*rchidaceae is the largest family of flowering plants, together with *Asteraceae,* with approximately 800 genera and more than 28,000 species, including almost 10% of all flowering plant species (1–3). The genus *Dendrobium* is the second largest genus after *Bulbophyllum* in the *Orchidaceae* and comprises over 1,100 species of epiphytic orchids, which are distributed across Asia, New Guinea, and Australia (4, 5). Although species of this genus are well known for their ornamental and medicinal value, many of them are threatened with extinction (5, 6).

The orchid roots are strongly associated with a wide variety of endophytic microbes (2, 5–7). The function of endophytic microbes of orchids, especially fungi, has attracted attention, and a number of studies have supported the idea that endophytic fungi play important roles in the growth and development of orchids throughout their life (6, 8–10). A prime example is that orchid seeds lack essential nutrients to maintain plant growth and thus depend on fungi for germination and carbon supply (6, 11). On the other hand, very little is known about the role of endophytic bacteria in the growth and development of orchids (7, 11). In general, endophytic bacteria play a crucial role in promoting plant growth and yield through nitrogen fixation, phytohormone production, nutrient acquisition, and biocontrol activities and also have the potential to create novel natural products like pharmaceutically relevant compounds (12–14). Therefore, understanding the interaction between *Dendrobium* plants and endophytic bacteria is a significant issue in order to develop new strategies for orchid protection and better utilization of its medicinal principles (2).

The predominant endophytic bacteria are likely to have profound effects on their hosts; thereby, analyzing their function may lead to a detailed understanding of plant-bacterium interactions. Recent advances in culture-independent methods, e.g., 16S rRNA gene amplicon sequencing, has made it possible to reveal the composition and diversity of the endophytic bacterial community of orchids (2, 7, 15). The metagenomic sequencing approach further allows prediction of their functional and metabolic potential, although to understand and verify the "true" function and/or discover unprecedented functions of endophytic bacteria, cultivation and isolation are ultimately essential. To date, however, no effective method of obtaining a pure culture of a wide variety of bacteria from the interior of plants (not just orchids) has been established. The fact that few environmental bacteria can grow in the laboratory (<1%) has been a limiting factor for unveiling the role of endophytic bacteria (7, 16).

Several simple methods for preparing media have been devised for widely isolating bacterial groups from environmental samples, including the use of gellan gum as a gelling reagent instead of agar (termed "GG" medium) (17) and separate autoclave sterilization of phosphate and agar (termed "PS" medium, where "S" represents "separately") (18). Both methods are very simple, but they have been found to improve the efficacy of isolation of diverse bacteria from several environmental samples, including soil, sediment, and/or freshwater. Furthermore, many reports have shown that these techniques are also effective for culturing recalcitrant bacteria inhabiting various environments (17–22). Nonetheless, these cultivation methods have not been adopted for obtaining axenic cultures of endophytic bacteria in plants, including orchids.

We hypothesized that these simply modified media (GG and PS) could be useful for isolating diverse indigenous endophytic bacteria of *Dendrobium* plants, including predominant and/or novel bacterial taxa. To verify this hypothesis, we isolated more than 700 endophytic bacteria from surface-sterilized *Dendrobium* roots using the different media and compared the phylogenetic compositions of the isolates with those of the endophytic bacterial community obtained from 16S rRNA gene amplicon sequencing. In addition, the present study further validated the effectiveness of the modified media for growing phylogenetically novel and/or recalcitrant endophytic isolates.

## RESULTS

**Analysis of endophytic bacterial communities of *Dendrobium* roots by 16S rRNA gene amplicon sequencing.** Endophytic bacterial communities of the two strains of *Dendrobium moniliforme* (green stem strain [GS] and white stem strain [WS])

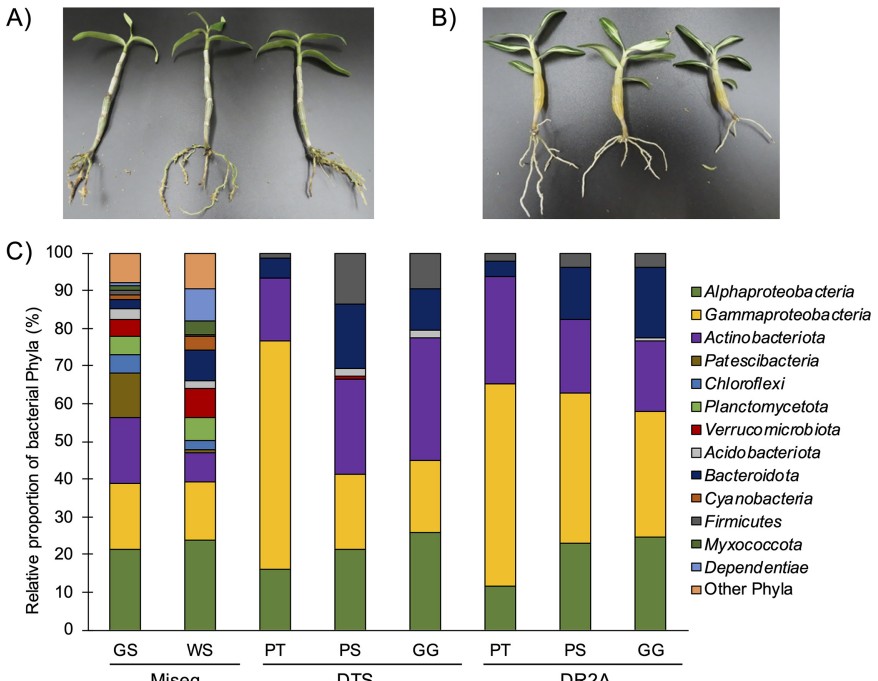

**FIG 1** Relative abundances of the endophytic bacterial phyla (or classes) in the roots of *Dendrobium* strains GS and WS according to MiSeq 16S rRNA gene amplicon sequencing and isolation using basal agar media (DTS and DR2A) prepared by the simply modified PS and GG methods and the conventional PT method. (A and B) Photographs of the roots of *Dendrobium* strains GS (A) and WS (B). (C) The phylum *Proteobacteria* is shown at the class level (*Alphaproteobacteria* and *Gammaproteobacteria*), and 15 additional phyla are included as "other phyla." The data from MiSeq 16S rRNA gene amplicon sequencing and isolation are shown for each *Dendrobium* strain (GS and WS) and each medium, respectively.

(Fig. 1A and B) were analyzed using 16S rRNA gene amplicon sequencing. The sequencing generated a total of 1,132,902 raw reads from the orchid root DNA samples (see Table S1 in the supplemental material). After merging of forward and reverse reads using divisive amplicon denoising algorithm 2 (DADA2) and removal of operational taxonomic units (OTUs) classified into chloroplasts, mitochondria, and archaea, the number of merged reads for each of the samples ranged from 43,557 to 51,299. The number of OTUs with 98% similarity varied from 772 to 934. The rarefaction curve indicated that the number of reads was sufficient to assess the diversity of the endophytic bacterial communities (Fig. S1).

A comparison of the endophytic bacterial community compositions at the phylum level showed that the following 8 of the top 10 abundant phyla were detected in both GS and WS: *Proteobacteria*, *Actinobacteriota*, *Verrucomicrobiota*, *Planctomycetota*, *Bacteroidota*, *Chloroflexi*, *Cyanobacteria*, and *Acidobacteriota* (Fig. 1C). The remaining 2 of the top 10 phyla of GS were *Patescibacteria* and *Firmicutes*, while those of WS were *Dependentiae* and *Myxococcota*. To clarify the bacterial groups dominating inside the *Dendrobium* roots, OTUs with a relative abundance of more than 1% were extracted as predominant OTUs (Fig. 2). Consequently, a total of 30 predominant OTUs were found, with 7 OTUs common to GS and WS, 11 OTUs specific to GS, and 12 OTUs specific to WS. These OTUs were classified into the following 9 phyla: *Proteobacteria* (11 OTUs), *Actinobacteriota* (8 OTUs), *Bacteroidota* (3 OTUs), *Chloroflexi* (2 OTUs), *Verrucomicrobiota* (2 OTUs), *Planctomycetota* (1 OTU), *Cyanobacteria* (1 OTU), *Patescibacteria* (1 OTU), and *Dependentiae* (1 OTU).

**Isolation efficacy of PT, PS, and GG media for endophytic bacteria of *Dendrobium* roots.** The isolation efficacy of diluted tryptic soy (DTS) and diluted R2A (DR2A) media prepared by the simply modified method of PS (separate autoclave sterilization of phosphate and agar) or GG (the use of gellan gum as a gelling reagent instead of agar) was compared with that of the conventional one (PT; the well-known conventional

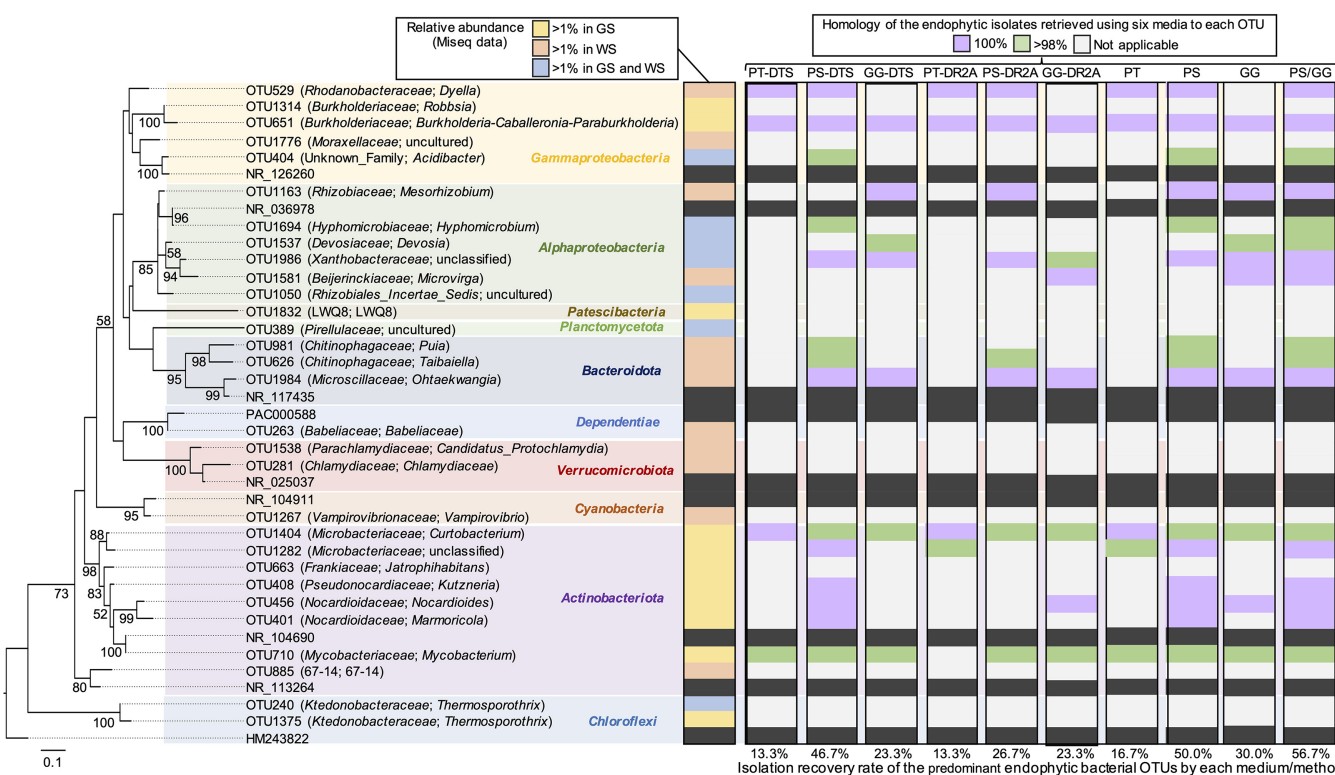

**FIG 2** Isolation recovery rate of predominant endophytic OTUs by each medium (PT-DTS, PS-DTS, GG-DTS, PT-DR2A, PS-DR2A, and GG-DR2A) or method (PT, PS, and GG). The predominant endophytic bacterial OTUs comprise more than 1.0% of relative abundance in the interior of the roots of either *Dendrobium* strain (GS or WS) according to MiSeq amplicon sequencing. Each OTU is followed by taxonomic information (family; genus), which was determined using the SILVA database (silva-138-99). The isolates obtained in this study which exhibit high homology (>98%) to the predominant endophytic bacterial OTUs are shown as the isolates affiliated with the predominant OTUs. The maximum likelihood phylogenetic tree of 16S rRNA gene V4 regions of the predominant endophytic bacterial OTUs was prepared using RAxML-NG (version 0.9.0) and Figtree (http://tree.bio.ed.ac.uk/software/figtree/). The accession number of each reference sequence is shown, and the sequence submitted under GenBank accession number HM243822 was used as an outgroup. ML bootstrap values were obtained using 100 bootstrap replicates and are shown on branches if above 50%.

method, i.e., autoclaving all medium nutrient components including phosphate and agar together). The numbers of endophytic bacterial isolates obtained from *Dendrobium* roots using six different media, i.e., PT-DTS, PS-DTS, GG-DTS, PT-DR2A, PS-DR2A, and GG-DR2A, were 151, 128, 114, 140, 117, and 99, respectively (Table 1). Based on the 16S rRNA gene sequencing analysis, a total of 749 isolates were classified into 97 "groups" using a 95% similarity cutoff value (corresponding to the genus level), which were designated instead of OTUs to avoid confusion with data from the MiSeq amplicon sequencing. Interestingly, regardless of the type of medium composition, the alpha-diversity indices (Simpson and Shannon) of PS- and GG-derived isolates were much higher than those of the PT isolates

**TABLE 1** Comparison of alpha-diversity values of the *Dendrobium* endophytic isolates obtained from PT, PS, and GG media[a]

| Medium | No. of isolates | No. of groups[a] | Alpha-diversity index[b] | |
|---|---|---|---|---|
| | | | Simpson | Shannon |
| PT-DTS | 151 | 29 | 0.77 | 2.21 |
| PS-DTS | 128 | 46 | 0.95 | 3.36 |
| GG-DTS | 114 | 41 | 0.94 | 3.24 |
| PT-DR2A | 140 | 26 | 0.79 | 2.22 |
| PS-DR2A | 117 | 41 | 0.94 | 3.19 |
| GG-DR2A | 99 | 34 | 0.94 | 3.14 |

[a]The bacterial isolates obtained in this study were classified with 95% similarity (corresponding to genus level) into 97 "groups."

[b]The indices of alpha-diversity based at the group level were averaged using 10 replicates of data rarefied to the lowest number of isolates obtained on a single medium.

(Table 1). Isolates belonging to the phyla *Actinobacteriota*, *Bacteroidota*, *Proteobacteria* (*Alphaproteobacteria* and *Gammaproteobacteria*), and *Firmicutes* were retrieved from all of the examined media (Fig. 1C). Intriguingly, the isolates of the phyla *Acidobacteriota* and *Verrucomicrobiota* were obtained only from the simply modified media (PS-DTS, GG-DTS, and/or GG-DR2A), not from the conventional ones (all the PT media) at all. The six isolates of the *Acidobacteriota* were classified into two groups: one is related to the genus *Edaphobacter* (1 isolate), and the other is related to the genus *Terriglobus* (5 isolates) (Table 2). An isolate of the *Verrucomicrobiota* was related to the genus *Chthoniobacter*. In addition, novel endophytic bacterial isolates (<95% similarity to the validly described strains) were successfully obtained only from PS and GG media (Table 2). All novel isolates were classified into 9 groups based on 98% similarity, and these groups were associated with either *Alphaproteobacteria*, *Gammaproteobacteria*, or *Bacteroidota*. Four or more groups of novel isolates were retrieved from any PS or GG medium. These results indicate that the PS and GG media were more effective for isolating a wide variety of *Dendrobium* root-inhabiting endophytic bacteria, including recalcitrant endophytic bacteria such as phylogenetically novel isolates and isolates of the rarely cultured phyla *Acidobacteriota* and *Verrucomicrobiota*.

We further verified how many of the predominant bacteria of the interior of the *Dendrobium* roots were retrieved with the media used in this study. For this, the sequence homology between all isolates and the predominant endophytic bacterial OTUs revealed by 16S rRNA gene amplicon sequencing was calculated using BLAST+ (23), and then the isolates exhibiting high homology (>98%) were selected as the isolates affiliated with the predominant OTUs. As a result, use of PS or GG medium allowed the isolation of the predominant bacterial groups more widely than did use of the corresponding PT medium (Fig. 2). The recovery rates of the predominant endophytic bacteria by PT-DTS, PS-DTS, GG-DTS, PT-DR2A, PS-DR2A, and GG-DR2A were 13.3%, 46.7%, 23.3%, 13.3%, 26.7%, and 23.3%, respectively. Surprisingly, the use of PS and GG media resulted in culturing and isolating of more than 50% (17 out of 30) of the predominant OTUs (Fig. 2). Besides, all predominant endophytic isolates obtained using PT media were also retrieved from PS and GG media. The 17 OTUs obtained here consisted of 8 OTUs belonging to *Proteobacteria* (72.7%), 6 OTUs belonging to *Actinobacteriota* (75%), and 3 OTUs belonging to *Bacteroidota* (100%), although the predominant OTUs of the phyla *Chloroflexi*, *Verrucomicrobiota*, *Planctomycetota*, *Cyanobacteria*, *Patescibacteria*, and *Dependentiae* were not obtained in this study. Notably, these predominant isolates included the novel endophytic bacterial isolates which were affiliated with OTU404 and OTU1984 (Table 2). Using PS media, we isolated members of the predominant bacterial OTUs more efficiently than when using GG media (Fig. 2). However, several predominant isolates (compatible with OTU1537 and OTU1581) were obtained only from GG media. Taken together, we found that high isolation efficacy of the predominant endophytic bacteria could be achieved by using both PS and GG media.

**Effect of PS and GG methods on growth of recalcitrant bacterial isolates.** To clarify a reason why the recalcitrant bacterial isolates such as the novel isolates (<95% similarity to valid strains) and *Acidobacteriota* and *Verrucomicrobiota* isolates were successfully retrieved with PS or GG media but not with PT media (Table 2), we further investigated the effect of PS and GG methods on colony formation of the endophytic recalcitrant bacterial isolates. Although all tested recalcitrant isolates formed colonies on even PT media, 25 of the novel isolates (63.0% of the total 46 isolates), 3 isolates affiliated with phylum *Acidobacteriota* (50.0% of the 6 isolates), and 1 isolate affiliated with phylum *Verrucomicrobiota* (100% of the 1 isolate) formed visible colonies on PS or GG plates at least twice as fast as they did on their corresponding PT plate (Fig. 3A). In particular, isolate GSA-72 of the phylum *Verrucomicrobiota* formed visible colonies on the PS-DTS plate (18 h of incubation) more than 8 times faster than it did on the PT-DTS plate (162 h of incubation) (Fig. 3B and C). Novel isolates belonging to the remaining seven groups, except for novel groups 2 and 3, were found to form colonies more

**TABLE 2** Recalcitrant endophytic bacterial isolates obtained on six different media from the interior of *Dendrobium* roots

| Recalcitrant bacterial isolates[a] | No. of isolates from: | | | | | | Closest-match valid strain[b] | | | | | Closest-match accession[c] | | Related predominant OTU (% similarity to OTUs)[d] |
|---|---|---|---|---|---|---|---|---|---|---|---|---|---|---|
| | PT-DTS | PS-DTS | GG-DTS | PT-DR2A | PS-DR2A | GG-DR2A | Representative isolate | Accession no. | Phylum/class | Genus | % Similarity | Accession no. (% similarity) | Source | |
| **Novel isolates** | | | | | | | | | | | | | | |
| Novel group 1 | 0 | 2 | 1 | 0 | 0 | 0 | WSC-37 | AB081581 | *Alphaproteobacteria* | *Rhizomicrobium* | 91.5 | PAC000263 (97.3) | Unknown | NA |
| Novel group 2 | 0 | 0 | 2 | 0 | 0 | 1 | GSC-63 | DQ672568 | *Alphaproteobacteria* | *Skermanella* | 91.5 | PAC000228 (98.3) | Unknown | NA |
| Novel group 3 | 0 | 1 | 0 | 0 | 0 | 0 | GSA-66 | JX412366 | *Gammaproteobacteria* | *Acidibacter* | 93.9 | FN554396 (97.9) | *Allium* rhizosphere | OTU404 (98.4) |
| Novel group 4 | 0 | 0 | 0 | 1 | 0 | 0 | GSB-61 | KM083135 | *Gammaproteobacteria* | *Sapientia* | 92.6 | JQ798403 (97.3) | Maize straw | NA |
| Novel group 5 | 0 | 0 | 1 | 0 | 0 | 0 | GSC-66 | JX412366 | *Gammaproteobacteria* | *Acidibacter* | 92.2 | PAC001319 (99.7) | Unknown | NA |
| Novel group 6 | 0 | 11 | 2 | 0 | 2 | 2 | WSA-37 | jgi.1048941 | *Bacteroidota* | *Ohtaekwangia* | 92.7 | AB240469 (93.6) | *Phragmites* rhizosphere | OTU1984 (100) |
| Novel group 7 | 0 | 1 | 3 | 0 | 1 | 3 | WSA-10 | JQ638910 | *Bacteroidota* | *Asinibacterium* | 92.9 | JN656858 (93.7) | Water | NA |
| Novel group 8 | 0 | 0 | 2 | 0 | 2 | 0 | GSC-53 | DQ244076 | *Bacteroidota* | *Niastella* | 94.9 | FJ479490 (98.6) | Grass | NA |
| Novel group 9 | 0 | 0 | 0 | 0 | 0 | 1 | GSD-26 | JX458466 | *Bacteroidota* | *Heliimonas* | 94.1 | JQ684312 (98.1) | Permafrost soil | NA |
| **Isolates affiliated with *Acidobacteriota*** | | | | | | | | | | | | | | |
| Group 1 | 0 | 3 | 1 | 0 | 0 | 1 | GSA-29 | CP003379 | *Acidobacteriota* | *Terriglobus* | 97.7 | JUGR01000001 (98.2) | Soil | NA |
| Group 2 | 0 | 0 | 1 | 0 | 0 | 0 | WSC-45 | KN050788 | *Acidobacteriota* | *Edaphobacter* | 99.2 | KN050788 (99.2) | Forest soil | NA |
| **Isolate affiliated with *Verrucomicrobiota*** | | | | | | | | | | | | | | |
| Group 1 | 0 | 1 | 0 | 0 | 0 | 0 | GSA-72 | ABVL01000001 | *Verrucomicrobiota* | *Chthoniobacter* | 96.2 | JF176805 (97.7) | Human skin | NA |

[a]Recalcitrant bacterial isolates include novel isolates whose sequences had less than 95% similarity to valid strains and isolates affiliated with the phyla *Acidobacteriota* and *Verrucomicrobiota*. These isolates were classified with 98% similarity into 9 groups of novel isolates, 2 groups of *Acidobacteriota*, and 1 *Verrucomicrobiota* group.

[b]Taxonomic identity of each group to valid strains was determined using the EzBioCloud server.

[c]Taxonomic identity of each group to all accessions was determined using the EzBioCloud server.

[d]Representative isolates with more than 98% similarity to the predominant endophytic bacterial OTUs, as described in Fig. 2, are shown. NA, not applicable.

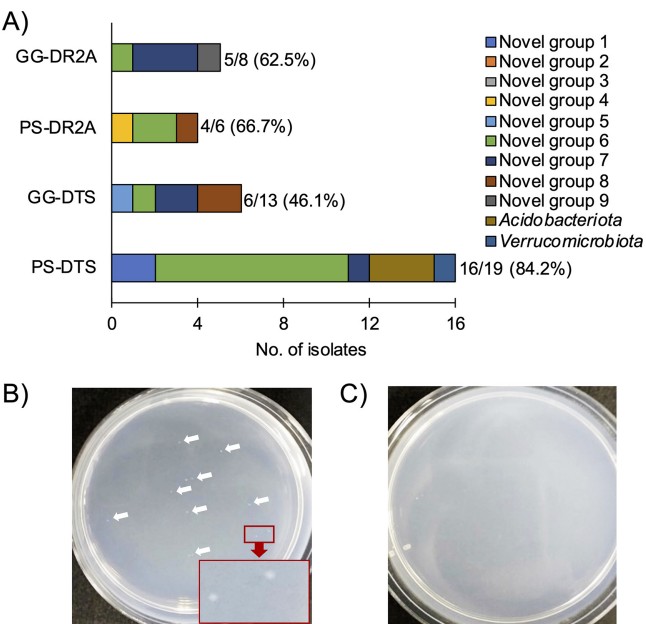

**FIG 3** Effect of PS and GG methods on visible colony formation of the phylogenetically novel endophytic isolates and *Acidobacteriota* and *Verrucomicrobiota* isolates. (A) Number of isolates showing at least two-times-faster colony formation on PS or GG media than that on their respective PT media. The novel group was defined as shown in Table 2. (B and C) Colony formation of isolate GSA-72 belonging to *Verrucomicrobiota* on the PS-DTS plate (B) or the PT-DTS plate (C) after 7 days of incubation. White arrows indicate the colonies.

rapidly on PS or GG plates than on any PT plate. These results suggest that such rapid colony formation on PS and GG plates is one of the plausible reasons for the successful isolation of these recalcitrant bacterial isolates.

## DISCUSSION

Cultivation and isolation of a wide range of endophytic bacteria including predominant ones are a critical issue that must be addressed to investigate their unidentified functions and biology, which may lead to the elucidation of the interactions of plants and endophytic bacteria. In the present study, we demonstrated that simply modified cultivation approaches (i.e., use of modified media and the improved recipe of the PS and GG methods, proposed in our previous studies) (17, 18) improved the isolation efficacy of even endophytic bacteria of *Dendrobium* roots. In fact, the alpha-diversity indices (Simpson and Shannon) of PS and GG medium isolates were much higher than those from their corresponding PT medium isolates (Table 1). Besides, the endophytic bacterial isolates obtained from *Dendrobium* roots using PS and GG media (PS-DTS, PS-DR2A, GG-DTS, and GG-DR2A) were distributed across six different phyla, including two rarely cultivated phyla, i.e., *Acidobacteriota* and *Verrucomicrobiota* (the other four phyla are *Actinobacteriota*, *Bacteroidota*, *Firmicutes*, and *Proteobacteria*), whereas those from media prepared by conventional methods (PT-DTS and PT-DR2A) belonged to only four phyla (*Actinobacteriota*, *Bacteroidota*, *Firmicutes*, and *Proteobacteria*) (Fig. 1C). So far, several studies have attempted to isolate endophytic bacteria of *Dendrobium* plants (7, 16, 24, 25). Wang and colleagues isolated endophytic bacteria from the interior of *Dendrobium* plants using a total of 11 different media (e.g., R2A agar, 10% nutrient agar, and humid acid agar) supplemented with 1% plant extracts of *Dendrobium* bodies (7). Despite such vigorous efforts, the isolated bacterial taxa were only three phyla (*Actinobacteriota*, *Firmicutes*, and *Proteobacteria*). Other studies have also isolated endophytic bacteria from inside *Dendrobium* bodies using media of nutrient agar, oatmeal agar, and/or ISP4 agar, but bacterial taxa other than the three phyla (*Actinobacteriota*, *Firmicutes*, and *Proteobacteria*) have not been isolated (16, 24, 25). In these previous studies, no

phylogenetically novel bacteria were obtained. Furthermore, using the PS and GG methods, we successfully isolated more than 50% (17 out of 30) of the predominant endophytic bacterial OTUs of *Dendrobium* roots, with a relative abundance of more than 1.0% revealed by MiSeq amplicon sequencing (Fig. 2). On the other hand, endophytic bacterial communities of *Dendorobium* roots revealed by MiSeq amplicon sequencing had higher diversity than those determined by culture-dependent methods (Fig. 1). The predominant OTUs belonging to the phyla *Chloroflexi*, *Verrucomicrobiota*, *Planctomycetota*, *Cyanobacteria*, *Patibacteria*, and *Dependentiae* were not isolated in this study (Fig. 2). These bacterial taxa are known to be difficult to isolate, and their isolation and culture are the next challenge. However, to our knowledge, the successful isolation of more than half is an unprecedented level of isolation efficiency for the dominant plant endophytic bacterial taxa. These results support the application of the PS and GG methods as effective for isolating a wide variety of endophytic bacteria, including more than 50% of the predominant ones from the interior of *Dendrobium* plants.

Previous studies including ours indicated that the application of PS and GG methods is effective for culturing recalcitrant bacteria in soil, sediment, sludge, and/or freshwater (17–21). In accordance with these reports, this study showed that a number of phylogenetically novel isolates (<95% similarity to valid strains) affiliated with the phyla *Proteobacteria* and *Bacteroidota*, and members of even rarely cultivated phyla *Acidobacteriota* and *Verrucomicrobiota* were successfully isolated with PS and/or GG media, whereas these bacteria were not obtained using PT media (Fig. 1C and Table 2). Notably, the isolates that were affiliated with novel groups 3 and 6 belonged to the predominant endophytic bacterial OTUs (OTU404 and OTU1984) (Table 2). Among nine novel groups, isolates of groups 3, 4, 6, and 8 are closely related to plant-derived 16S rRNA gene sequences (Table 2). This suggests that our novel isolates are likely plant associated (not only with orchids but also with other plant species). Such success in isolating the predominant endophytic bacteria might be the result of the growth promotion of the modified cultivation methods adopted in this study. Indeed, the majority of the recalcitrant endophytic bacterial isolates formed visible colonies on PS or GG plates faster than they did on their respective PT plates (Fig. 3A). In particular, visible colony formation of the isolate of the phylum *Verrucomicrobiota* was found to be at least eight times faster on the PS plate than on the PT plate (Fig. 3B and C). Similarly, our previous studies reported that the recalcitrant bacterial isolates obtained from soil, sediment, and freshwater grew only on, or better on, PS or GG plates (18–20). For instance, the isolates of the rarely cultivated phylum *Gemmatimonadota* showed colony formation on PS or GG plates that was dramatically different from that on the corresponding PT plates (18, 20). Collectively, it was suggested that the application of the PS and GG methods might enable cultivation and isolation of recalcitrant bacterial isolates not only from soil, sediment, sludge, and freshwater but also from associated plants (e.g., endophytes) by facilitating their colony formations.

To the best of our knowledge, we first succeeded in obtaining bacterial isolates of the phyla *Acidobacteriota* and *Verrucomicrobiota* from the interior of orchid plants (Fig. 1C). In particular, *Verrucomicrobiota* have been rarely isolated from not only the interior of orchids but also from the interior of whole plants. Only two isolates (both belong to subdivision 4, out of seven subdivisions proposed for phylum *Verrucomicrobiota*) recovered from the root endosphere of *Oryza sativa* and *Oryza longistaminata* were reported (26). The isolate obtained in this study belongs to subdivision 2 (the closest relative is the genus *Chthoniobacter*) and is phylogenetically different from the above-mentioned two isolates (26). This isolate together with the previous ones would be useful for elucidating functional roles of *Verrucomicrobiota* in plant bodies that have long been largely unknown. The phylum *Acidobacteriota* is composed of diverse members spanning 26 subdivisions and has recently attracted much attention due to its members that are associated with soil-plant ecosystems worldwide (27). For instance, recent studies showed that the nonendophytic isolates affiliated with subdivisions 1, 3, and 6 of phylum *Acidobacteriota* possessed growth-promoting effects on plants such as *Arabidopsis thaliana* and duckweed species

(28, 29). Since the six isolates obtained in this study belong to subdivision 1 and are related to the genus *Edaphobacter* or *Terriglobus*, perhaps these isolates might contribute to the growth and development of orchid plants. Accordingly, the endophytic bacterial isolates affiliated with the phyla *Verrucomicrobiota* and *Acidobacteriota* obtained in the present study must be valuable resources for comprehensive functional analysis of the endophytes of these two phyla.

In conclusion, the present study demonstrated that the DTS and DR2A media prepared by the PS and GG methods enable isolation of endophytic bacteria from the interior of *Dendrobium* roots as follows: (i) more than 50% of the predominant endophytic bacterial taxa of *Dendrobium* roots, (ii) some phylogenetically novel isolates (<95% similarity to valid strains) affiliated with the phyla *Proteobacteria* and *Bacteroidota*, and (iii) bacteria affiliated with the rarely cultivated phyla *Verrucomicrobiota* and *Acidobacteriota* were successfully obtained. These results suggest that the PS and GG methods are effective for widely isolating endophytic bacteria from the interior of plant bodies, in addition to soil, sediment, and/or freshwater (17, 18, 20). Future studies revealing the functions of the endophytic bacterial isolates obtained in this study would contribute to unveiling the unknown roles of endophytic bacteria in the growth and development of *Dendrobium* plants and perhaps may shed light on developing new strategies for their protection and better utilization of their medicinal principles.

## MATERIALS AND METHODS

***Dendrobium* root sampling.** This study used two strains of *Dendrobium moniliforme*, which were kindly provided by Hiroshi Noda and Hiroko Noda. The *Dendrobium* strains characteristically had green stems (GS strain) and white stems (WS strain), respectively (Fig. 1A and B), and were cultivated on sphagnum moss under natural light with regular watering for more than 2 years. Root sections were collected from three plants of each strain and then immediately surface sterilized by the following methods. The roots were treated with 75% ethanol for 30 s, 1% (vol/vol) Tween 20 for 1 min, 3% sodium hypochlorite for 10 min, and 75% ethanol for 30 s and then were rinsed with sterile distilled water (SDW) three times and cut into ca. 0.5-cm-long sections with a sterile scalpel. To confirm whether the sterilization process was successful, roots were rolled on the plates of six different media used for the bacterial isolation experiment as described below and also 100 $\mu$L of the final water rinse was inoculated and spread on the same plates, which consistently yielded no bacterial colonies incubated at 25°C for 4 weeks. For the isolation of endophytic bacteria, 1 g of roots was homogenized gently in 9 mL of SDW using a mortar and pestle. To extract DNA from surface-sterilized roots, 0.5 g of roots was homogenized in liquid nitrogen using a mortar and pestle, and the homogenate was immediately stored at −80°C until use.

**Endophytic bacterial community analysis by 16S rRNA gene amplicon sequencing.** 16S rRNA gene amplicon sequencing was performed to determine the predominant endophytic bacterial taxa of the roots of both strains of *Dendrobium* (GS and WS). Genomic DNA was extracted from each homogenized root using a FastDNA spin kit for soil (MP Biomedicals, CA, USA) according to the manufacturer's protocol. DNA extraction was repeated three times. Amplifications of the V4 region of 16S rRNA genes were performed using a primer set specific for the V4 region (505F, GTGCCAGCMGCCGCGGTAA; 806R, GGACTACHVGGGTWTCTAAT). *Ex Taq* DNA polymerase (TaKaRa Bio, Shiga, Japan) was used for PCR amplification, and the thermal cycling was performed with a denaturation step at 94°C for 2 min, followed by 23 cycles at 94°C for 30 s, 50°C for 30 s, and 72°C for 30 s, and a final extension step at 72°C for 5 min. The 16S rRNA gene amplicon libraries were paired-end sequenced on an Illumina MiSeq platform using 2 × 250 bp overlapping paired-end reads (Illumina, CA, USA). Sequence processing was conducted using the QIIME 2 pipeline (version 2019.7). The paired-end fastq files were demultiplexed with demux-summarize and then were processed by quality filtering, merging of the paired ends, and chimera removal with divisive amplicon denoising algorithm 2 (DADA2) (30). In DADA2 processing, we used options to do the following: (i) remove primer sequences, (ii) truncate forward and reverse reads to 194 bp and 125 bp, respectively, and (iii) truncate the reads containing the base with a quality score less than or equal to 15. Each read was clustered into operational taxonomic units (OTUs) at 98% similarity using VSEARCH (31). Taxonomy was assigned to each OTU using the SILVA database (silva-138-99) using feature classifier. Subsequently, reads classified into chloroplasts, mitochondria, and archaea were removed. OTUs with a relative abundance of more than 1.0% in the interior of either GS or WS roots were defined as the predominant endophytic bacterial OTUs. Predominant bacterial OTUs were further identified using the EzBioCloud server (32), and the corresponding OTUs were excluded when there were no hits. Furthermore, a phylogenetic tree of the predominant endophytic bacterial OTUs was constructed using maximum likelihood (ML) methods. Sequences were aligned using MAFFT with default settings, and then an ML tree was constructed using RAxML-NG (version 0.9.0) with 100 bootstrap replicates (33). Tree results were viewed using Figtree (version 1.4.4) (http://tree.bio.ed.ac.uk/software/figtree/).

**Isolation of endophytic bacteria from *Dendrobium* roots.** To isolate endophytic bacteria from *Dendrobium* roots, two types of basal agar media supplemented with the fungicide cycloheximide (50 $\mu$g mL$^{-1}$) were used: diluted R2A medium (DR2A; yeast extract, 0.05 g L$^{-1}$; peptone, 0.05 g L$^{-1}$;

dextrose, 0.05 g L$^{-1}$; starch, 0.05 g L$^{-1}$; Casamino Acids, 0.05 g L$^{-1}$; dipotassium phosphate, 0.3 g L$^{-1}$; magnesium sulfate heptahydrate, 49.2 mg L$^{-1}$; sodium pyruvate, 0.3 g L$^{-1}$; agar, 15 g L$^{-1}$) and diluted tryptic soy medium (DTS: tryptone, 0.17 g L$^{-1}$; Soytone, 0.03 g L$^{-1}$; dextrose, 0.025 g L$^{-1}$; sodium chloride, 0.05 g L$^{-1}$; dipotassium phosphate, 0.025 g L$^{-1}$; agar, 15 g L$^{-1}$). Both basal media were prepared by the simply modified methods (PS and GG) and by the conventional PT method (autoclaving all medium nutrient components including phosphate and agar together). In the PS method, all medium nutrient components, including phosphate and agar, were separately autoclaved and mixed. In the GG method, gellan gum was used instead of agar as a gelling agent and CaCl$_2$ was added at a final concentration of 3 mM. A 100-$\mu$L aliquot of each serial dilution of the *Dendrobium* root suspension was spread onto the surface of each plate (90 mm in diameter) in triplicate. Each plate was incubated at 25°C in the dark for 21 to 28 days. After incubation, colonies appearing on each plate were randomly selected and streaked using quadrant streaking on fresh plates for further purification.

**Phylogenetic analysis of the endophytic isolates.** The genomic DNA of each isolate was prepared using an InstaGene matrix (Bio-Rad Laboratories, Hercules, CA, USA) according to the manufacturer's instructions. The 16S rRNA gene of each isolate was amplified with 27f (AGAGTTTGATCMTGGCTCAG) and 1492r (GGYTACCTTGTTACGACTT) primers and PrimeSTAR HS DNA polymerase (TaKaRa Bio, Shiga, Japan). The amplification conditions were 30 cycles at 98°C for 10 s, 55°C for 5 s, and 72°C for 90 s. The PCR products were purified using ExoSAP-IT express PCR cleanup reagents (Thermo Fisher Scientific, Inc., Japan). Cycle sequencing was performed using the 907r (CCGTCAATTCMTTTRAGTTT) primer with the BigDye Terminator v3.1 cycle sequencing kit (Thermo Fisher Scientific, Inc.) according to the manufacturer's instructions. The fluorescent labeled fragments were purified using the BigDye XTerminator purification kit (Thermo Fisher Scientific) and were analyzed by an ABI 3730xl DNA analyzer (Thermo Fisher Scientific). To compare the isolation efficacies of the media, the sequences obtained were assigned to OTUs by using the CD-HIT-EST program (34) with a cutoff value of 95% (corresponding to the genus level) (35), and alpha-diversity indices (Shannon and Simpson) were calculated using the vegan 2.5-6 package in R 3.6.1 software. To avoid confusion with data from the MiSeq amplicon sequencing, the OTUs of the isolates were designated as a group. The partial 16S rRNA gene sequences of each group were identified using the EzBioCloud server. The isolates obtained in this study that exhibited high homology (>98%) to the predominant endophytic bacterial OTUs are shown as the isolates affiliated with the predominant OTUs.

**Effect of PS and GG methods on growth of recalcitrant endophytic bacterial isolates.** In order to clarify some of reasons why the recalcitrant endophytic bacterial isolates, including the phylogenetically novel isolates (<95% similarity to valid strains based on their partial 16S rRNA gene sequence) and the isolates affiliated with members of even rarely cultivated phyla (e.g., *Acidobacteriota* and *Verrucomicrobiota*), were obtained from PS or GG media but not from PT media, the effects of the PS and GG methods on colony formation of the recalcitrant isolates were examined based on the criterion of at least two-times-faster visible colony formation on PS or GG plates than on their respective PT plates, as described in our previous report (20). Each isolate was precultured on the medium plate used for the isolation experiment and then suspended with SDW. Each suspension was spread onto the surface of the medium plate used for the preculture and the corresponding PT medium and then incubated at 25°C under dark conditions. The colony formation was monitored every 18 h by the naked eye and finally by using a stereomicroscope.

**Data availability.** Sequence data were deposited in the Sequence Read Archive database under accession numbers DRR354706 to DRR354711.

## SUPPLEMENTAL MATERIAL

Supplemental material is available online only.
**SUPPLEMENTAL FILE 1**, PDF file, 0.1 MB.

## ACKNOWLEDGMENTS

We thank Hiroshi Noda and Hiroko Noda for providing two strains of *Dendrobium moniliforme*.

This work was supported mainly by Grant-in-Aids for Young Scientists (grant no. 20K15643), Scientific Research on the Innovative Area "Post-Koch Ecology" (MEXT KAKENHI no. JP19H05683), and JSPS Fellows (grant no. 19J01859) from the Ministry of Education, Culture, Sports, Science and Technology of Japan.

We declare no conflicts of interest.

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
