## [Reviewer comments · Microbiology Spectrum]

Microbiology Spectrum

Improved cultivation and isolation of diverse endophytic bacteria inhabiting *Dendrobium* roots by using simply modified agar media

Tomoki Nishioka and Hideyuki Tamaki

Corresponding Author(s): Tomoki Nishioka, National Institute of Advanced and Industrial Science and Technology

Review Timeline:

Submission Date:	June 14, 2022
Editorial Decision:	August 16, 2022
Revision Received:	August 18, 2022
Accepted:	September 30, 2022

Editor: Courtney Robinson

Reviewer(s): Disclosure of reviewer identity is with reference to reviewer comments included in decision letter(s). The following individuals involved in review of your submission have agreed to reveal their identity: Gisela Manuela de França Bettencourt (Reviewer #1); Madhusmita Borah (Reviewer #2)

Transaction Report:

DOI: <https://doi.org/10.1128/spectrum.02238-22>

August 16, 2022

Dr. Tomoki Nishioka
National Institute of Advanced and Industrial Science and Technology
Bioproduction Research Institute
1-1-1 Higashi
Tsukuba, Ibaraki 3058566
Japan

Re: Spectrum02238-22 (Improved cultivation and isolation of diverse endophytic bacteria inhabiting *Dendrobium* roots by using simply modified agar media)

Dear Dr. Tomoki Nishioka:

Thank you for submitting your manuscript to Microbiology Spectrum. As you will see your paper is very close to acceptance! Please modify the manuscript along the lines the reviewers have recommended. As these revisions are quite minor, I expect that you should be able to turn in the revised paper in less than 30 days, if not sooner. You will find the reviewers' comments below.

When submitting the revised version of your paper, please provide (1) point-by-point responses to the issues I raised in your cover letter, and (2) a PDF file that indicates the changes from the original submission (by highlighting or underlining the changes) as file type "Marked Up Manuscript - For Review Only". Please use this link to submit your revised manuscript. Detailed instructions on submitting your revised paper are below.

Link Not Available

Sincerely,

Courtney Robinson

Reviewer comments:

Reviewer #1 (Comments for the Author):

Well done for the article and the investigation. Your results showed that we need to seek for new strategies to overcome some difficulties in cultivating recalcitrant species.
Just need to point on line 308 is written dada2, I supposed was to be data.

Reviewer #2 (Comments for the Author):

Comments

1. Line 32: "...contribute to clarify their unknown role in the growth and development of *Dendrobium* plants." Please justify the above statement as the study did not deal with the role of endophytes in the growth and development of *Dendrobium* plants.
2. Was the endophytic diversity also analyzed from other parts of the plant except roots?
3. Line 218: "These results support that the....." The findings shows that application of PS and GG media is effective in isolating more than 50% of the predominant endophytic bacteria in *Dendrobium*. Give an explanation for the effectiveness of the

aforesaid media.

4. Line 230: "These suggests that our novel isolates....." Give reference
5. Line 331: instead of "all nutrients" please give details on experimental methods and be precise.
6. Line 251: add space in "abovementioned"
7. Line 524: Remove the comma after "isolates"

Suggestions

The discussion section should explain a comparison of the diversity of endophytes observed in Dendrobium plants from culture-dependent and culture-independent approach.

Preparing Revision Guidelines

- point-by-point responses to the issues I raised in your cover letter
- Upload a compare copy of the manuscript (without figures) as a "Marked-Up Manuscript" file.
- Each figure must be uploaded as a separate file, and any multipanel figures must be assembled into one file.
- Manuscript: A .DOC version of the revised manuscript
- Figures: Editable, high-resolution, individual figure files are required at revision, TIFF or EPS files are preferred

Please return the manuscript within 60 days; if you cannot complete the modification within this time period, please contact me. If you do not wish to modify the manuscript and prefer to submit it to another journal, please notify me of your decision immediately so that the manuscript may be formally withdrawn from consideration by Microbiology Spectrum.

[revised manuscript text omitted]

interactions. Recent advances in culture-independent methods, e.g. 16S rRNA gene amplicon
sequencing, has made it possible to reveal the composition and diversity of endophytic bacterial
community of orchids (2, 7, 15). Metagenomic sequencing approach further allows to predict
their functional and metabolic potential. Even though, to understand and verify “true” function
and/or discover unprecedented functions of the endophytic bacteria, cultivation and isolation is
ultimately essential. To date, however, no effective method obtaining pure culture of a wide
variety of bacteria from the interior of plants (not just orchids) has been established. The fact that
few environmental bacteria can grow in the laboratory (<1%) has been a limiting factor for
unveiling the role of endophytic bacteria (7, 16).

Several simple methods for preparing media have been devised for widely isolating
bacterial groups from the environmental samples, which includes the use of gellan gum as a
gelling reagent instead of agar (termed “GG” medium) (17) and separate autoclave sterilization
of phosphate and agar (termed “PS” medium, where “S” represents “separately”) (18). Both
methods are very simple, but those have been found to improve the efficacy of diverse bacterial
isolation from several environmental samples including soil, sediment and/or freshwater.
Furthermore, many reports showed that these techniques are also effective for culturing
recalcitrant bacteria inhabiting various environments (17–22). Nonetheless, these cultivation
methods have not been adopted for obtaining the axenic cultures of endophytic bacteria in plants
including orchid.

We hypothesized that these simple modified media (GG and PS) could be useful for
isolating diverse indigenous endophytic bacteria of *Dendrobium* plants, including predominant
and/or novel bacterial taxa. To verify this hypothesis, we isolated more than 700 endophytic
bacteria from the surface-sterilized *Dendrobium* roots using the different media and compared
the phylogenetic compositions of the isolates with those of endophytic bacterial community
obtained from 16S rRNA gene amplicon sequencing. Besides, the present study further validated
the effectiveness of the modified media for growing the phylogenetically novel and/or recalcitrant
endophytic isolates.

**Results**

***Dendrobium* roots-endophytic bacterial community analysis by 16S rRNA gene amplicon** 102 **sequencing**

Endophytic bacterial communities of the two strains of *Dendrobium moniliforme* (“GS”;
green stem strain and “WS”; white stem strain; Fig. 1A and B) were analyzed using 16S rRNA
gene amplicon sequencing. The sequencing generated a total of 1,132,902 raw reads from the
orchid root DNA samples (Supplementary Table S1). After merging forward and reverse reads
using dada2 and removing OTUs classified into chloroplasts, mitochondria, and archaea, the

[revised manuscript text omitted]

Although all tested recalcitrant isolates formed colonies on even PT media, 25 of novel isolates
(63.0% of the total 46 isolates), 3 isolates affiliated with phylum *Acidobacteriota* (50.0% of the
6 isolates), and 1 isolate affiliated with phylum *Verrucomicrobiota* (100% of the 1 isolate) formed
visible colonies on PS or GG plates at least twice faster than on their corresponding PT plate (Fig.
3A). In particular, the isolate GSA-72 of the phylum *Verrucomicrobiota* formed visible colonies
on PS-DTS plate (18 h incubation) more than eighth as fast as on PT-DTS plate (162 h incubation)
(Fig. 3B and C). Novel isolates belonging to the remaining seven groups, except for novel groups
2 and 3, were found to form colonies more rapidly on PS or GG plates than on each PT plate.
These results suggested that such a rapid colony formation on PS and GG plates is one of the
plausible reasons for successful isolation of these recalcitrant bacterial isolates.

190190

Discussion

Cultivation and isolation of a wide range of endophytic bacteria including predominant
ones is a critical issue that must be addressed to investigate their unidentified functions and
biology which may lead to elucidation of plant-endophytic bacteria interactions. In the present
study, we demonstrated that the simply modified cultivation approaches (i.e., a use of modified
media and the improved recipe, PS and GG methods, proposed by our previous studies) (17, 18)
improved the isolation efficacy of even endophytic bacteria of *Dendrobium* roots. In fact, the
alpha diversity indices (Simpson and Shannon) of PS- and GG-isolates were much higher than
those from their corresponding PT-isolates (Table 1). Besides, the endophytic bacterial isolates
obtained from *Dendrobium* roots using PS and GG media (PS-DTS, PS-DR2A, GG-DTS, and
GG-DR2A) were distributed across six different phyla including two rarely cultivated phyla such
as *Acidobacteriota* and *Verrucomicrobiota* (other four phyla are *Actinobacteriota*, *Bacteroidota*,
*Firmicutes*, and *Proteobacteria*), whereas those from media prepared by conventional methods
(PT-DTS and PT-DR2A) belonged to only four phyla (*Actinobacteriota*, *Bacteroidota*,
*Firmicutes*, and *Proteobacteria*) (Fig. 1C). So far, several studies have attempted to isolate
endophytic bacteria of *Dendrobium* plants (7, 16, 24, 25). Wang and colleagues isolated
endophytic bacteria from the interior of *Dendrobium* plant using a total of 11 different media (e.g.,
R2A agar, 10% nutrient agar, and humid acid agar) supplemented with 1% plant extracts of
*Dendrobium* bodies (7). Despite such vigorous efforts, the isolated bacterial taxa were only three
phyla (*Actinobacteriota*, *Firmicutes*, and *Proteobacteria*). Another studies have also isolated
endophytic bacteria from inside *Dendrobium* bodies using media of nutrient agar, oatmeal agar,
and/or ISP4 agar, but bacterial taxa other than the three phyla (*Actinobacteriota*, *Firmicutes*, and
*Proteobacteria*) have not been isolated (16, 24, 25). In these previous studies, no phylogenetically
novel bacteria were obtained. Furthermore, using PS and GG methods, we successfully isolated
more than 50% (17 out of 30) of the predominant endophytic bacterial OTUs of *Dendrobium*

roots with a relative abundance of more than 1.0% revealed by the MiSeq amplicon sequencing
(Fig. 2). To our knowledge, this is an unprecedented level of isolation efficacy of predominant
plant endophytic bacterial taxa. These results support that the application of the PS and GG
methods is effective for isolating a wide variety of endophytic bacteria including predominant
ones from the interior of *Dendrobium* plants.

[revised manuscript text omitted]

In conclusion, the present study demonstrated that the DTS and DR2A media prepared
by PS and GG methods enables the endophytic bacteria isolation from the interior of *Dendrobium*
roots as follows: i) more than 50% of the predominant endophytic bacterial taxa of *Dendrobium*
roots, ii) some phylogenetically novel isolates (<95% similarity to valid strains) affiliated with
phyla *Proteobacteria* and *Bacteroidota*, and iii) bacteria affiliated with the rarely cultivated phyla
*Verrucomicrobiota* and *Acidobacteriota* were successfully obtained. These results suggest that
the PS and GG methods is effective for widely isolating endophytic bacteria from interior of plant
bodies, in addition to soil, sediment and/or freshwater (17, 18, 20). Future studies revealing
functions of the endophytic bacterial isolates obtained in this study would contribute to unveiling
the unknown roles of endophytic bacteria in the growth and development of *Dendrobium* plants,
and perhaps may shed light on developing new strategies for their protection and better utilization
of their medicinal principles.

275275

**Materials and methods**

***Dendrobium* roots sampling**

This study used two strains of *Dendrobium moniliforme*, which had been kindly
provided by Mr. Hiroshi Noda and Ms. Hiroko Noda. They characteristically had green stem
("GS" strain) and white stem ("WS" strain), respectively (Fig. 1A and B), and were cultivated on
sphagnum moss under natural light with regular watering for more than two years. Root sections
were collected from three plants of each strain and then immediately surface-sterilized as
following methods. The roots were treated with 75% ethanol for 30 s; 1 % (v/v) tween 20 for 1
284 min; 3% sodium hypochlorite for 10 min; 75% ethanol for 30 s, and then were rinsed with sterile
distilled water (SDW) three times and cut into ca. 0.5 cm long sections with a sterile scalpel. To
confirm whether the sterilization process was successful, roots were rolled on the plates of six
different media used for the bacterial isolation experiment as described below, and also 100 μ L

of the final water rinse was inoculated and spread on the same plates, which consistently yielded
no bacterial colonies incubated at 25 °C for four weeks. For the isolation of endophytic bacteria,
one gram of roots was homogenized gently in 9 ml of SDW using a mortar and pestle. To extract
DNA from surface-sterilized roots, 0.5 gram of roots was homogenized in liquid nitrogen using
a mortar and pestle, and the homogenate was immediately stored at –80°C until use.

293293

Endophytic bacterial community analysis by 16S rRNA gene amplicon sequencing

A 16S rRNA gene amplicon sequencing was performed to determine predominant
endophytic bacterial taxa of both *Dendrobium* (GS and WS) roots. Genomic DNA was extracted
from each homogenized root using FastDNA SPIN Kit for soil (MP Biomedicals, CA, USA)
according to the manufacturer's protocol. The DNA extraction was repeated three times.
Amplifications of V4 region of 16S rRNA genes were performed using primer set specific for V4
region (505F: GTGCCAGCMGCCGCGGTAA; and 806R: GGACTACHVGGGTWTCTAAT).
Ex Taq[®] DNA polymerase (Takara Bio, Shiga, Japan) was used for PCR amplification and the
thermal cycle step was performed with a denaturation step at 94°C for 2 min, followed by 23
cycles at 94°C for 30 s, 50°C for 30 s, and 72°C for 30 s and a final extension step at 72°C for 5
304 min. The 16S rRNA gene amplicon libraries were paired-end sequenced on an Illumina MiSeq
platform using 2 × 250 bp overlapping paired-end reads (Illumina, CA, USA). Sequence
processing was conducted using Qiime2 pipeline (version 2019.7). The paired-end fastq files
were demultiplexed with demux-summarize and then were processed by quality filtering, merging
of the paired ends and chimera removal with dada2 (30). In dada2 processing, we used options to
do the following, 1) primer sequences were removed, 2) forward and reverse reads were truncated
to 194 bp and 125 bp, respectively, and 3) the reads containing the base with quality score less
than or equal to 15 were truncated. Each read was clustered into operational taxonomic units
(OTUs) at 98% similarity using VSEARCH (31). Taxonomy was assigned to each OTU using the
SILVA database (silva-138-99) using feature-classifier. Subsequently, reads classified into
chloroplasts, mitochondria, and archaea were removed. OTUs with a relative abundance of more
than 1.0% either in the interior of GS or WS root were defined as the predominant endophytic
bacterial OTUs. Predominant bacterial OTUs were further identified using EzBioCloud server
(32), and the corresponding OTUs were excluded when there were no hits. Furthermore, a
phylogenetic tree of the predominant endophytic bacterial OTUs was constructed using maximum
likelihood (ML) methods. Sequences were aligned using MAFFT with default settings and then
a ML tree was constructed using RAxML-NG (version 0.9.0) with 100 bootstrap replicates (33).
Tree results were viewed using Figtree (version 1.4.4) (<http://tree.bio.ed.ac.uk/software/figtree/>).

322322

Isolation of endophytic bacteria from *Dendrobium* roots

To isolate endophytic bacteria from *Dendrobium* roots, two types of basal agar media
supplemented with the fungicide cycloheximide ($50 \mu\text{g mL}^{-1}$) were used: diluted R2A (DR2A;
yeast extract 0.05 g L^{-1} , peptone 0.05 g L^{-1} , dextrose 0.05 g L^{-1} , starch 0.05 g L^{-1} , casamino acids
0.05 g L^{-1} , dipotassium phosphate 0.3 g L^{-1} , magnesium sulfate heptahydrate 49.2 mg L^{-1} , sodium
pyruvate 0.3 g L^{-1} , agar 15 g L^{-1}) and diluted tryptic soy (DTS: tryptone 0.17 g L^{-1} , soytone 0.03
329 g L^{-1} , dextrose 0.025 g L^{-1} , sodium chloride 0.05 g L^{-1} , dipotassium phosphate 0.025 g L^{-1} , agar
15 g L^{-1}). Both basal media were prepared by simple modified methods ("PS" and "GG"), and
conventional "PT" (autoclaving all nutrients including phosphate and agar together) method. In
the PS method, all nutrients including phosphate and agar were separately autoclaved and mixed.
In the GG method, gellan gum is used instead of agar as a gelling agent and CaCl_2 was added at
a final concentration of 3 mM. A 100- μL aliquot of each serial dilution of the *Dendrobium* roots
suspension was spread onto the surface of each plate (90 mm in diameter) in triplicate. Each plate
was incubated at 25°C in the dark for 21–28 days. After incubation, colonies appeared on each
plate were randomly selected and streaked using quadrant streaking on fresh plates for further
purification.

**Phylogenetic analysis of the endophytic isolates**

The genomic DNA of each isolate was prepared using InstaGene matrix (Bio-Rad
Laboratories, Hercules, CA, USA) according to the manufacturer's instructions. The 16S rRNA
gene of each isolate was amplified with 27f (AGAGTTTGATCMTGGCTCAG) and 1492r
(GGYTACCTTGTTACGACTT) primers and PrimeSTAR HS DNA polymerase (Takara Bio,
Shiga, Japan). The amplification conditions were 30 cycles at 98°C for 10 s, at 55°C for 5 s, and
at 72°C for 90 s. The PCR products were purified using the ExoSAP-IT Express PCR Cleanup
Reagents (Thermo Fisher Scientific Inc., Japan). Cycle sequencing was performed using 907r
(CCGTCAATTCMTTTRAGTTT) primer with the BigDyeTM Terminator v3.1 Cycle Sequencing
Kit (Thermo Fisher Scientific Inc.) according to manufacturer's instructions. The fluorescent-
labeled fragments were purified using the BigDye XTerminatorTM Purification Kit (Thermo
Fisher Scientific) and were analyzed by an ABI 3730xl DNA Analyzer (Thermo Fisher Scientific).
To compare the isolation efficacy of each medium, the sequences obtained were assigned to OTUs
by using the CD-HIT-EST program (34) with a cutoff value of 95% (corresponding to genus
level) (35), and alpha-indices (shannon and simpson) were calculated using the vegan 2.5-6
package in R 3.6.1 software. To avoid confusion with data from the Miseq amplicon sequencing,
the OTU of the isolate was designated as group. The partial 16S rRNA gene sequences of each
group were identified using a EzBioCloud server. The isolates obtained in this study which exhibit
high homology ($>98\%$) to the predominant endophytic bacterial OTUs were shown as the isolates
affiliated with the predominant OTUs.

Effect of PS and GG methods on growth of recalcitrant endophytic bacterial isolates

In order to clarify part of reasons why the recalcitrant endophytic bacterial isolates
including the phylogenetically novel isolates (<95% similarity to valid strains based on their
partial 16S rRNA gene sequence) and the isolates affiliated with members of even rarely
cultivated phyla (e.g. *Acidobacteriota* and *Verrucomicrobiota*) were obtained from PS or GG
media but not from PT media, the effect of PS and GG methods on the colony formation of the
recalcitrant isolates were examined based on the criteria of at least twice faster visible colony
formation on PS or GG plates than on their respective PT plate, as described by our previous
report (20). Each isolate was precultured on the medium plate used for the isolation experiment
and then suspended with SDW. Each suspension was spread onto the surface of the medium plate
used for the pre-culture and the corresponding PT medium, and then incubated at 25°C under dark
conditions. The colony formation was monitored every 18 h by the naked eye and finally by using
stereomicroscope.

374374

Data availability

Sequence data were deposited in the Sequence Read Archive database under accession
numbers DRR354706–DRR354711.

Declaration of competing interest

The authors declare no conflicts of interest.

381381

Acknowledgments

[revised manuscript text omitted]

 *Dendrobium* strains (“GS”; green stem strain, and “WS”; white stem strain) according to the
 Miseq 16S rRNA gene amplicon sequencing and isolation using basal agar media (“DTS ”;
 diluted tryptic soy, and “DR2A”; diluted R2A) prepared by simple modified methods (“PS”;
 separate sterilization of phosphate and agar, and “GG”; use of gellan gum as a gelling reagent)
 and conventional “PT” (autoclaving the phosphate and agar together) method. A and B.
 Photographs of the roots of *Dendrobium* strains GS (A) and WS (B). C. Phylum *Proteobacteria*
 was shown at class level (*Alphaproteobacteria* and *Gammaproteobacteria*) and additional 15
 phyla are included as other phyla. The data from Miseq 16S rRNA gene amplicon sequencing and
 isolation were shown for each *Dendrobium* strain (GS and WS) and each medium, respectively.

Figure 2. The legend is shown in the next page.

**Figure 2.** Isolation recovery rate of predominant endophytic OTUs by each medium (PT-DTS,
PS-DTS, GG-DTS, PT-DR2A, PS-DR2A, and GG-DR2A) or method (PT, PS, and GG). The
predominant endophytic bacterial OTUs comprise more than 1.0% of relative abundance either
in the interior of roots of *Dendrobium* strains (GS or WS) according to the Miseq amplicon
sequencing. Each OTU is followed by taxonomy (Family; Genus) which was determined using
the SILVA database (silva-138-99). The isolates obtained in this study which exhibit high
homology (>98%) to the predominant endophytic bacterial OTUs were shown as the isolates
affiliated with the predominant OTUs. Maximum likelihood phylogenetic tree of 16S rRNA genes
V4 regions of the predominant endophytic bacterial OTUs was prepared using RAxML-NG
(version 0.9.0) and Figtree (<http://tree.bio.ed.ac.uk/software/figtree/>). The accession number of
each reference sequence was shown and the accession number HM243822 was used as an
outgroup. ML bootstrap values were obtained using 100 bootstrap replicates and are shown on
branches if above 50%.

521

522

**Figure 3.** Effect of PS and GG methods on visible colony formation of the phylogenetically novel
 endophytic isolates, and *Acidobacteriota* and *Verrucomicrobiota* isolates. (A) The number of the
 isolates showing at least twice faster colony formation on PS or GG media than their respective
 PT medium. Novel group was defined as described in Table 2. Colony formation of the isolate
 GSA-72 belonging to *Verrucomicrobiota* on the PS-DTS plate (B) or PT-DTS plate (C) after 7
 528 days incubation. White arrows indicate the colonies.

529

530

5315 **Table 1.** Comparison of alpha diversity values of the *Dendrobium* endophytic isolates obtained
 3 from PT, PS, and GG media
 1

5325

Medium	No. of isolates	No. of groups ^a	α -diversity indices ^b	
			Simpson	Shannon
PT-DTS	151	29	0.77	2.21
PS-DTS	128	46	0.95	3.36
GG-DTS	114	41	0.94	3.24
PT-DR2A	140	26	0.79	2.22
PS-DR2A	117	41	0.94	3.19
GG-DR2A	99	34	0.94	3.14

^a The bacterial isolates obtained in this study were classified with 95% similarity (corresponding to genus level) into 97 "groups".

^b The indices of α -diversity based at group level were averaged using 10 replicates of data rarefied to the lowest number of isolates obtained on a single medium.

5345

3

4

5355

3

5

5365

3

6

5375

3

7

5385

3

8

**Table 2.** Recalcitrant endophytic bacterial isolates obtained on the six different media from the interior of *Dendrobium* roots.

Recalcitrant bacterial isolates ^a	The No. of isolates from						Representative isolate	Closest match valid strain ^b			Closest match accession ^c			Related predominant OTUs (Similarity to OTUs) ^d
	PT-DTS	PS-DTS	GG-DTS	PT-DR2A	PS-DR2A	GG-DR2A		Accession	Phylum/Class	Genus	Similarity	Accession (Similarity)	Source	
Novel isolates														
Novel group 1	0	2	1	0	0	0	WSC-37	AB081581	Alphaproteobacteria Rhizomicrobium	91.5%	PAC000263 (97.3%)	Unknown	Not applicable	
Novel group 2	0	0	2	0	0	1	GSC-63	DQ672568	Alphaproteobacteria Skermanella	91.5%	PAC000228 (98.3%)	Unknown	Not applicable	
Novel group 3	0	1	0	0	0	0	GSA-66	JX412366	Gammaproteobacteria Acidibacter	93.9%	FN554396 (97.9%)	Allium rhizosphere	OTU404 (98.4%)	
Novel group 4	0	0	0	0	1	0	GSB-61	KM083135	Gammaproteobacteria Sapientia	92.6%	JQ798403 (97.3%)	Maize straw	Not applicable	
Novel group 5	0	0	1	0	0	0	GSC-66	JX412366	Gammaproteobacteria Acidibacter	92.2%	PAC001319 (99.7%)	Unknown	Not applicable	
Novel group 6	0	11	2	0	2	2	WSA-37	jgi.1048941	Bacteroidota Ohtaekwangia	92.7%	AB240469 (93.6%)	Phragmites rhizosphere	OTU1984 (100%)	
Novel group 7	0	1	3	0	1	3	WSA-10	JQ638910	Bacteroidota Asinibacterium	92.9%	JN656858 (93.7%)	Water	Not applicable	
Novel group 8	0	0	2	0	2	0	GSC-53	DQ244076	Bacteroidota Niastella	94.9%	FJ479490 (98.6%)	Grass	Not applicable	
Novel group 9	0	0	0	0	0	1	GSD-26	JX458466	Bacteroidota Heliomonas	94.1%	JQ684312 (98.1%)	Permafrost soil	Not applicable	
Isolates affiliated with Acidobacteriota														
Group 1	0	3	1	0	0	1	GSA-29	CP003379	Acidobacteriota Terriglobus	97.7%	JUGR01000001 (98.2)	Soil	Not applicable	
Group 2	0	0	1	0	0	0	WSC-45	KN050788	Acidobacteriota Edaphobacter	99.2%	KN050788 (99.2%)	Forest soil	Not applicable	
Isolate affiliated with Verrucomicrobiota														
Group 1	0	1	0	0	0	0	GSA-72	ABVL01000001	Verrucomicrobiota Chthoniobacter	96.2%	JF176805 (97.7%)	Human skin	Not applicable	

540

541

542 ^a Recalcitrant bacterial isolates include novel isolates whose sequences had less than 95% similarity to valid strains and isolates affiliated with phyla
 *Acidobacteriota* and *Verrucomicrobiota*. These isolates were classified with 98% similarity into 9 groups of novel isolates, 2 groups of *Acidobacteriota*,
 and 1 *Verrucomicrobiota* group.

545 ^b Taxonomic identity of each group to valid strains was determined using the EzBioCloudserver.

546 ^c Taxonomic identity of each group to all accessions was determined using the EzBioCloudserver.

547 ^d Representative isolates with more than 98% similarity to the predominant endophytic bacterial OTUs, as described in Fig. 2, were shown.

548

549

5505 **Supplementary Figure and Table**

5

0

5515

5

1

5525

5

2

5535

5

3

**Supplementary Figure S1.** Rarefaction curves for endophytic bacterial operational taxonomic
units (OTUs) from each *Dendrobium* sample. GS: green stem-*Dendrobium* root and WS: white
stem-*Dendrobium* root.

557

**Supplementary Table S1** Statistics of Miseq amplicon sequencing of endophytic bacterial
communities in the *Dendrobium* roots

Root sample	No. of raw reads	No. of reads after Qiime2 process	No. of OTUs
GS-1	183219	45725	843
GS-2	200196	51299	934
GS-3	179852	43557	837
WS-1	189484	46691	787
WS-2	183392	43614	772
WS-3	196759	50086	792

GS: green stem-*Dendrobium* root and WS: white stem-*Dendrobium* root.

563

August 18, 2022

Dear Dr. Courtney Robinson,

Thank you very much for serving as the editor for our manuscript (Manuscript #: Spectrum02238-22) and giving us the opportunity to revise it. We also greatly appreciate the careful review and constructive comments from two reviewers. We would like to submit the revised version of manuscript. According to the constructive comments, we have thoroughly revised our manuscript and addressed all the issues raised. We truly believe that we have improved the quality of the manuscript to meet the journal's standards of *Microbiology Spectrum*. For more information, please confirm the responses to all the comments following this letter.

We hope the revised manuscript is now suitable for publication in *Microbiology Spectrum*.

Yours sincerely,

Tomoki Nishioka and Hideyuki Tamaki
Bioproduction Research Institute,
National Institute of Advanced and Industrial Science and Technology (AIST),
1-1-1 Higashi, Tsukuba, Ibaraki 305-8566, Japan
Phone No: +81 29 861 6591, Fax No: +81 29 861 6587
E-mail: tomoki-nishioka@aist.go.jp (T. Nishioka)
tamaki-hideyuki@aist.go.jp (H. Tamaki)

Point-by-point responses to Reviewer 1

Reviewer's comments in black

Responses by authors in green

Well done for the article and the investigation. Your results showed that we need to seek for new strategies to overcome some difficulties in cultivating recalcitrant species.

Just need to point on line 308 is written dada2, I supposed was to be data.

----- We greatly appreciate your positive evaluation. We sincerely apologize for having caused confusion. “dada2” is an error-corrected clustering method, namely Divisive Amplicon Denoising Algorithm 2. Also, dada2 should have been written as DADA2. Therefore, we have changed “dada2 “to “divisive amplicon denoising algorithm 2 (DADA2)”.

(Lines 109, 323, and 324 in the revised manuscript).

Point-by-point responses to Reviewer 2

Reviewer's comments in black

Responses by authors in green

We deeply appreciate your valuable comments. According to the suggestions, we have revised the manuscript as described below.

Comments

1. Line 32: "...contribute to clarify their unknown role in the growth and development of *Dendrobium* plants." Please justify the above statement as the study did not deal with the role of endophytes in the growth and development of *Dendrobium* plants.

----- We agree with this comment. As you pointed out, this study did not deal with the role of endophytes in the growth and development of *Dendrobium* plants. Here, we just intended to mention that the obtained endophytes should be useful living materials for future study clarifying the unknown roles in growth and development of *Dendrobium* plants, but our description seems to mislead the readers as you suggested. To clarify this point, we have revised this sentence as follows.

"...makes it possible to conduct future studies to clarify their unknown roles associated with growth of *Dendrobium* plants."

(Line 32-33 in the revised manuscript).

2. Was the endophytic diversity also analyzed from other parts of the plant except roots?

----- In this study, we did not analyze the endophytic diversity of other parts of the plant except roots. As the future study, we would like to investigate endophytes in other parts by both culture-dependent and culture-independent approaches.

3. Line 218: "These results support that the....." The findings shows that application of PS and GG media is effective in isolating more than 50% of the predominant endophytic bacteria in *Dendrobium*. Give an explanation for the effectiveness of the aforesaid media.

----- As suggested, we have revised the sentence as follows:

"These results support that the application of the PS and GG methods is effective for isolating a wide variety of endophytic bacteria including more than 50% of predominant ones from the interior of *Dendrobium* plants. "

(Line 230-232 in the revised manuscript).

4. Line 230: "These suggests that our novel isolates....." Give reference

----- As suggested, we have added "(Table 2)" for reference.

(Line 242 in the revised manuscript).

5. Line 331: instead of "all nutrients" please give details on experimental methods and be precise.

----- As suggested, we have changed "all nutrients" to "all medium nutrient components".

(Lines 131, 348, and 349 in the revised manuscript).

6. Line 251: add space in "abovementioned"

----- Corrected.

7. Line 524: Remove the comma after "isolates"

----- Corrected.

Suggestions

The discussion section should explain a comparison of the diversity of endophytes observed in *Dendrobium* plants from culture-dependent and culture-independent approach.

----- As suggested, we have added and revised the sentences as follows:

Original sentence:

To our knowledge, this is an unprecedented level of isolation efficacy of predominant plant endophytic bacterial taxa.

↓

Revised sentences:

On the other hand, endophytic bacterial communities of *Dendrobium* roots revealed by MiSeq amplicon sequencing were higher diversity than those by culture dependent methods (Fig. 1). The predominant OTUs belonging to phyla *Chloroflexi*, *Verrucomicrobiota*, *Planctomycetota*, *Cyanobacteria*, *Patibacteria*, and *Dependentiae* were not isolated in this study (Fig. 2). These bacterial groups are known to be difficult to isolate, and their isolation and culture will be the next challenge. However, to our knowledge, the successful isolation of more than half is an unprecedented level of isolation efficiency for the dominant plant endophytic bacterial taxa.

(Line 222-229 in the revised manuscript).

September 30, 2022

Dr. Tomoki Nishioka
National Institute of Advanced and Industrial Science and Technology
Bioproduction Research Institute
1-1-1 Higashi
Tsukuba, Ibaraki 3058566
Japan

Re: Spectrum02238-22R1 (Improved cultivation and isolation of diverse endophytic bacteria inhabiting *Dendrobium* roots by using simply modified agar media)

Dear Dr. Tomoki Nishioka:

Your manuscript has been accepted, and I am forwarding it to the ASM Journals Department for publication. You will be notified when your proofs are ready to be viewed.

Sincerely,

Courtney Robinson
Editor, Microbiology Spectrum